# Finetuning Offline World Models in the Real World

**Yunhai Feng**[1*] **Nicklas Hansen**[1*] **Ziyan Xiong**[12*]
**Chandramouli Rajagopalan**[1] **Xiaolong Wang**[1]

[1]University of California San Diego  [2]Tsinghua University

**Abstract:** Reinforcement Learning (RL) is notoriously data-inefficient, which makes training on a real robot difficult. While model-based RL algorithms (*world models*) improve data-efficiency to some extent, they still require hours or days of interaction to learn skills. Recently, offline RL has been proposed as a framework for training RL policies on *pre-existing* datasets without any online interaction. However, constraining an algorithm to a fixed dataset induces a state-action distribution shift between training and inference, and limits its applicability to new tasks. In this work, we seek to get the best of both worlds: we consider the problem of pretraining a world model with offline data collected on a real robot, and then finetuning the model on online data collected by planning with the learned model. To mitigate extrapolation errors during online interaction, we propose to regularize the planner *at test-time* by balancing estimated returns and (epistemic) model uncertainty. We evaluate our method on a variety of visuo-motor control tasks in simulation and on a real robot, and find that our method enables few-shot finetuning to seen and unseen tasks even when offline data is limited. Videos are available at https://yunhaifeng.com/FOWM.

**Keywords:** Model-Based Reinforcement Learning, Real-World Robotics

## 1  Introduction

Reinforcement Learning (RL) has the potential to train physical robots to perform complex tasks autonomously by interacting with the environment and receiving supervisory feedback in the form of rewards. However, RL algorithms are notoriously data-inefficient and require large amounts (often millions or even billions) of online environment interactions to learn skills due to limited supervision [1, 2, 3]. This makes training on a *real* robot difficult. To circumvent the issue, prior work commonly rely on custom-built simulators [4, 5, 1] or human teleoperation [6, 7] for behavior learning, both of which are difficult to scale due to the enormous cost and engineering involved. Additionally, these solutions each introduce additional technical challenges such as the simulation-to-real gap [8, 1, 9, 10] and the inability to improve over human operators [11, 12], respectively.

Recently, offline RL has been proposed as a framework for training RL policies from *pre-existing* interaction datasets without the need for online data collection [13, 14, 15, 16, 17, 18, 19, 20]. Leveraging existing datasets alleviates the problem of data-inefficiency without suffering from the aforementioned limitations. However, any pre-existing dataset will invariably not cover the entire state-action space, which leads to (potentially severe) extrapolation errors, and consequently forces algorithms to learn overly conservative policies [21, 22, 23, 24]. We argue that extrapolation errors are less of an issue in an online RL setting, since the ability to collect new data provides an intrinsic self-calibration mechanism: by executing overestimated actions and receiving (comparably) negative feedback, value estimations can be adjusted accordingly.

In this work, we seek to get the best of both worlds. We consider the problem of pretraining an RL policy on pre-existing interaction data, and subsequently finetuning said policy on a limited amount

---

*Equal contribution. Correspondence to Yunhai Feng<yuf020@ucsd.edu>.

7th Conference on Robot Learning (CoRL 2023), Atlanta, USA.

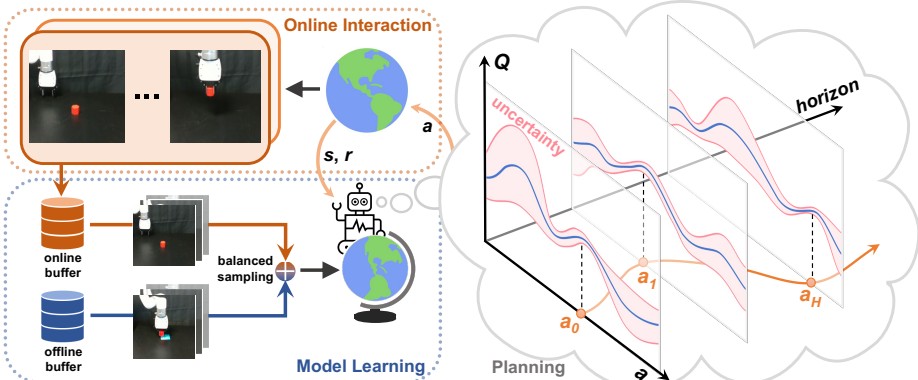

*Figure 1.* **Approach.** We propose a framework for offline pretraining and online finetuning of world models directly in the real world, without reliance on simulators or synthetic data. Our method iteratively collects new data by **planning with the learned model**, and **finetunes the model** on a combination of pre-existing data and newly collected data. Our method can be finetuned few-shot on unseen task variations in $\leq$**20 trials** by leveraging novel *test-time* regularization during planning.

of data collected by online interaction. Because we consider only a very limited amount of online interactions ($\leq$**20 trials**), we shift our attention to model-based RL (MBRL) [25] and the TD-MPC [26] algorithm in particular, due to its data-efficient learning. However, as our experiments will reveal, MBRL alone does not suffice in an offline-to-online finetuning setting when data is scarce. In particular, we find that planning suffers from extrapolation errors when queried on unseen state-action pairs, and consequently fails to converge. While this issue is reminiscent of overestimation errors in model-free algorithms, MBRL algorithms that leverage planning have an intriguing property: their (planning) policy is non-parametric and can optimize arbitrary objectives *without* any gradient updates. Motivated by this key insight, we propose a framework for offline-to-online finetuning of MBRL agents (*world models*) that mitigates extrapolation errors in planning via novel *test-time* behavior regularization based on (epistemic) model uncertainty. Notably, this regularizer can be applied to both purely offline world models *and* during finetuning.

We evaluate our method on a variety of continuous control tasks in simulation, as well as visuo-motor control tasks on a real xArm robot. We find that our method outperforms state-of-the-art methods for offline and online RL across most tasks, and enables few-shot finetuning to *unseen* tasks and task variations even when offline data is limited. For example, our method improves the success rate of an offline world model **from 22% to 67% in just *20 trials*** for a real-world visual *pick* task with *unseen* distractors. We are, to the best of our knowledge, the first work to investigate offline-to-online finetuning with MBRL on real robots, and hope that our encouraging few-shot results will inspire further research in this direction.

## 2 Preliminaries: Reinforcement Learning and the TD-MPC Algorithm

We start by introducing our problem setting and MBRL algorithm of choice, TD-MPC, which together form the basis for the technical discussion of our approach in Section 3.

**Reinforcement Learning** We consider the problem of learning a visuo-motor control policy by interaction, formalized by the standard RL framework for infinite-horizon Partially Observable Markov Decision Processes (POMDPs) [27]. Concretely, we aim to learn a policy $\pi_\theta \colon \mathcal{S} \times \mathcal{A} \mapsto \mathbb{R}_+$ that outputs a conditional probability distribution over actions $\mathbf{a} \in \mathcal{A}$ conditioned on a state $\mathbf{s} \in \mathcal{S}$ that maximizes the expected return (cumulative reward) $\mathcal{R} = \mathbb{E}_{\pi_\theta} \left[ \sum_{t=0}^{\infty} \gamma^t r_t \right]$, where $t$ is a discrete time step, $r_t$ is the reward received by executing action $\mathbf{a}_t$ in state $\mathbf{s}_t$ at time $t$, and $\gamma \in [0, 1)$ is a discount factor. We leverage an MBRL algorithm in practice, which decomposes $\pi_\theta$ into multiple learnable components (a *world model*), and uses the learned model for planning. For brevity, we use subscript $\theta$ to symbolize learnable parameters throughout this work. In a POMDP, environment interactions obey an (unknown) transition function $\mathcal{T} \colon \mathcal{S} \times \mathcal{A} \mapsto \mathcal{S}$, where states $\mathbf{s}$ themselves are as-

sumed unobservable. However, we can define approximate environment states $\mathbf{s} \doteq (\mathbf{o}_1, \mathbf{o}_2, \ldots, \mathbf{o}_n)$ from sensory observations $\mathbf{o}_{1:n}$ obtained from *e.g.* cameras or robot proprioceptive information.

**TD-MPC**  Our work extends TD-MPC [26], an MBRL algorithm that plans using Model Predictive Control (MPC) with a world model and terminal value function that is jointly learned via Temporal Difference (TD) learning. TD-MPC has two intriguing properties that make it particularly relevant to our setting: *(i)* it uses planning, which allows us to regularize action selection at test-time, and *(ii)* it is lightweight relative to other MBRL algorithms, which allows us to run it real-time. We summarize the architecture in Figure 2. Concretely, TD-MPC learns five components: *(1)* a representation $\mathbf{z} = h_\theta(\mathbf{s})$ that maps high-dimensional inputs $\mathbf{s}$ to a compact latent representation $\mathbf{z}$, *(2)* a latent dynamics model $\mathbf{z}' = d_\theta(\mathbf{z}, \mathbf{a})$ that predicts the latent representation at the next timestep, and three prediction heads: *(3)* a reward predictor $\hat{r} = R_\theta(\mathbf{z}, \mathbf{a})$ that predicts the instantaneous reward, *(4)* a terminal value function $\hat{q} = Q_\theta(\mathbf{z}, \mathbf{a})$, and *(5)* a latent policy guide $\hat{\mathbf{a}} = \pi_\theta(\mathbf{z})$ that is used as a behavioral prior for planning. We use $\mathbf{z}', \mathbf{s}'$ to denote the successor (latent) states of $\mathbf{z}, \mathbf{s}$ in a subsequence, and use $\hat{\mathbf{a}}, \hat{r}, \hat{q}$ to dif-

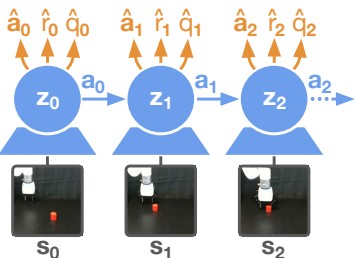

*Figure 2.* **Architecture.** Our world model encodes an observation $\mathbf{s}_0$ into its latent **representation** $\mathbf{z}_0$, and then recurrently predicts future latent states $\mathbf{z}_{1:h}$ as well as optimal **actions** $\hat{\mathbf{a}}_{0:h}$, **rewards** $\hat{r}_{0:h}$, and **values** $\hat{q}_{0:h}$. Future states $\mathbf{s}_{1:h}$ provide supervision for learning but are not required for planning.

ferentiate predictions from observed (ground-truth) quantities $\mathbf{a}, r, q$. In its original formulation, TD-MPC is an online off-policy RL algorithm that maintains a replay buffer $\mathcal{B}$ of interactions, and jointly optimizes all components by minimizing the objective

$$\mathcal{L}(\theta) = \mathbb{E}_{(\mathbf{s}, \mathbf{a}, r, \mathbf{s}')_{0:h} \sim \mathcal{B}} \left[ \sum_{t=0}^{h} \left( \underbrace{\|\mathbf{z}'_t - \text{sg}(h_\phi(\mathbf{s}'_t))\|_2^2}_{\text{Latent dynamics}} + \underbrace{\|\hat{r}_t - r_t\|_2^2}_{\text{Reward}} + \underbrace{\|\hat{q}_t - q_t\|_2^2}_{\text{Value}} - \underbrace{Q_\theta(\mathbf{z}_t, \hat{\mathbf{a}}_t)}_{\text{Action}} \right) \right] \quad (1)$$

where $(\mathbf{s}, \mathbf{a}, r, \mathbf{s}')_{0:h}$ is a subsequence of length $h$ sampled from the replay buffer, $\phi$ is an exponentially moving average of $\theta$, sg is the `stop-grad` operator, $q_t = r_t + \gamma Q_\phi(\mathbf{z}'_t, \pi_\theta(\mathbf{z}'_t))$ is the TD-target, and gradients of the last term (action) are taken only wrt. the policy parameters. Constant coefficients balancing the losses are omitted. We refer to Hansen et al. [26] for additional implementation details, and instead focus our discussion on details pertaining to our contributions. During inference, TD-MPC plans actions using a sampling-based planner (MPPI) [28] that iteratively fits a time-dependent multivariate Gaussian with diagonal covariance over the space of action sequences such that return – as evaluated by simulating actions with the learned model – is maximized. For a (latent) state $\mathbf{z}_0 = h_\theta(\mathbf{s}_0)$ and sampled action sequence $\mathbf{a}_{0:h}$, the estimated return $\hat{\mathcal{R}}$ is given by

$$\hat{\mathcal{R}} = \gamma^h \underbrace{Q_\theta(\mathbf{z}_h, \mathbf{a}_h)}_{\text{Value}} + \sum_{t=0}^{h-1} \gamma^t \underbrace{R_\theta(\mathbf{z}_t, \mathbf{a}_t)}_{\text{Reward}}, \quad \mathbf{z}_{t+1} = \underbrace{d_\theta(\mathbf{z}_t, \mathbf{a}_t)}_{\text{Latent dynamics}}, \quad \mathbf{z}_0 = \underbrace{h_\theta(\mathbf{s}_0)}_{\text{Encoder}} . \quad (2)$$

To improve the rate of convergence in planning, a fraction of sampled action sequences are generated by the learned policy $\pi_\theta$, effectively inducing a behavioral prior over possible action sequences. While $\pi_\theta$ is implemented as a deterministic policy, Gaussian action noise can be injected for stochasticity. TD-MPC has demonstrated excellent data-efficiency in an online RL setting, but suffers from extrapolation errors when naïvely applied to our problem setting, which we discuss in Section 3.

## 3  Approach: A *Test-Time* Regularized World Model and Planner

We propose a framework for offline-to-online finetuning of world models that mitigates extrapolation errors in the model via novel *test-time* regularization during planning. Our framework is summarized in Figure 1, and consists of two stages: *(1)* an *offline* stage where a world model is pretrained on pre-existing offline data, and *(2)* an *online* stage where the learned model is subsequently

finetuned on a limited amount of online interaction data. While we use TD-MPC [26] as our back-bone world model and planner, our approach is broadly applicable to any MBRL algorithm that uses planning. We start by outlining the key source of model extrapolation errors when used for offline RL, then introduce our test-time regularizer, and conclude the section with additional techniques that we empirically find helpful for the few-shot finetuning of world models.

## 3.1 Extrapolation Errors in World Models Trained by Offline RL

All methods suffer from extrapolation errors when trained on offline data and evaluated on unseen data due to a state-action distribution shift between the two datasets. In this context, value overestimation in model-free $Q$-learning methods is the most well-understood type of error [14, 16, 17, 19]. However, MBRL algorithms like TD-MPC face unique challenges in an offline setting: state-action distribution shifts are present not only in value estimation, but also in (latent) dynamics and reward prediction when estimating the return of sampled trajectories as in Equation 2. We will first address value overestimation, and then jointly address other types of extrapolation errors in Section 3.2.

Inspired by Implicit $Q$-learning (IQL; [19]), we choose to mitigate the overestimation issue by applying TD-backups only on *in-sample* actions. Specifically, consider the value term in Equation 1 that computes a TD-target $q$ by querying $Q_\phi$ on a latent state $\mathbf{z}'_t$ and potentially out-of-sample action $\pi_\theta(\mathbf{z}'_t)$. To avoid out-of-sample actions in the TD-target, we introduce a state-conditional value estimator $V_\theta$ and reformulate the TD-target as $q_t = r_t + \gamma V_\theta(\mathbf{z}'_t)$. This estimator can be optimized by an asymmetric $\ell_2$-loss (expectile regression):

$$\mathcal{L}_V(\theta) = |\tau - \mathbb{1}_{\{Q_\phi(\mathbf{z}_t, \mathbf{a}_t) - V_\theta(\mathbf{z}_t) < 0\}}|(Q_\phi(\mathbf{z}_t, \mathbf{a}_t) - V_\theta(\mathbf{z}_t))^2, \tag{3}$$

where $\tau \in (0, 1)$ is a constant *expectile*. Intuitively, we approximate the maximization in $V_\theta(\mathbf{z}_t) = \max_{\mathbf{a}_t} Q_\phi(\mathbf{z}_t, \mathbf{a}_t)$ for $\tau \to 1$, and are increasingly conservative for smaller $\tau$. Note that $\mathbf{a}_t$ is the action from the dataset (replay buffer), and thus no out-of-sample actions are needed. For the same purpose of avoiding out-of-sample actions, we replace the action term for policy learning in Equation 1 by an advantage weighted regression (AWR) [29, 30, 21] loss $\exp(\beta(Q_\phi(\mathbf{z}_t, \mathbf{a}_t) - V_\theta(\mathbf{z}_t))) \log \pi_\theta(\mathbf{a}_t | \mathbf{z}_t)$, where $\beta \geq 0$ is a temperature parameter.

## 3.2 Uncertainty Estimation as Test-Time Behavior Regularization

While only applying TD-backups on in-sample actions is effective at mitigating value overestimation during offline *training*, the world model (including dynamics, reward predictor, and value function) may still be queried on unseen state-action pairs during *planning*, *i.e.*, when estimating returns using Equation 2. This can result in severe extrapolation errors despite a cautiously learned value function. To address this additional source of errors, we propose a *test-time* behavior regularization technique that balances estimated returns and (epistemic) model uncertainty when evaluating Equation 2 for sampled action sequences. By regularizing estimated returns, we retain the expressiveness of planning with a world model despite imperfect state-action coverage. Avoiding actions for which the outcome is highly uncertain is strongly desired in an offline RL setting where no additional data can be collected, but conservative policies likewise limit exploration in an online RL setting. Unlike prior offline RL methods that predominantly learn an explicitly and consistently conservative value function and/or policy, regularizing planning based on model uncertainty has an intriguing property: as planning continues to cautiously explore and the model is finetuned on new data, the epistemic uncertainty naturally decreases. This makes test-time regularization based on model uncertainty highly suitable for both few-shot finetuning and continued finetuning over many trajectories.

To obtain a good proxy for epistemic model uncertainty [31] with *minimal* computational over-head and architectural changes, we propose to utilize a small ensemble of value functions $Q_\theta^{(1)}, Q_\theta^{(2)}, \ldots, Q_\theta^{(N)}$ similar to Chen et al. [32]. We optimize the $Q$-functions with TD-targets discussed in Section 3.1, and use a random subset of (target) $Q$-networks to estimate the $Q$-values in Equation 3. Although ensembling all components of the world model may yield a better estimate of epistemic uncertainty, our choice of ensembling is lightweight enough to run on a real robot and is empirically a sufficiently good proxy for uncertainty. We argue that with a $Q$-ensemble, we can

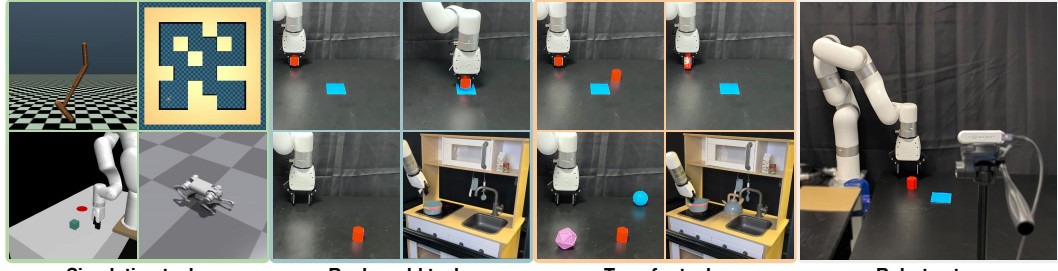

**Simulation tasks**     **Real-world tasks**     **Transfer tasks**     **Robot setup**

*Figure 3.* **Tasks.** We consider diverse tasks in simulation and on a real robot. Our real-world tasks use raw pixels as input. Our method achieves high success rates in offline-to-online transfer to both seen and unseen tasks in just 20 online trials on a real robot.

*Table 1.* **Real-world offline-to-online results.** Success rate (%) as a function of online finetuning trials. Mean of 18 trials and 2 seeds.

| | | online | offline-to-online | |
|---|---|---|---|---|
| | Trials | TD-MPC | TD-MPC | **Ours** |
| **Reach** | 0 | $0_{\pm 0}$ | $50_{\pm 18}$ | $\mathbf{72_{\pm 6}}$ |
| | 10 | $0_{\pm 0}$ | $67_{\pm 12}$ | $\mathbf{94_{\pm 6}}$ |
| | 20 | $0_{\pm 0}$ | $\mathbf{78_{\pm 12}}$ | $89_{\pm 0}$ |
| **Pick** | 0 | $0_{\pm 0}$ | $0_{\pm 0}$ | $0_{\pm 0}$ |
| | 10 | $0_{\pm 0}$ | $28_{\pm 6}$ | $\mathbf{33_{\pm 0}}$ |
| | 20 | $0_{\pm 0}$ | $28_{\pm 6}$ | $\mathbf{50_{\pm 6}}$ |
| **Kitchen** | 0 | $0_{\pm 0}$ | $0_{\pm 0}$ | $\mathbf{11_{\pm 11}}$ |
| | 10 | $0_{\pm 0}$ | $33_{\pm 11}$ | $\mathbf{56_{\pm 11}}$ |
| | 20 | $0_{\pm 0}$ | $61_{\pm 17}$ | $\mathbf{78_{\pm 0}}$ |

*Table 2.* **Finetuning to unseen real-world tasks.** Success rate (%) of our method for each task variation shown in Figure 3. We include 4 successful transfers and 1 failure. See Appendix B for task descriptions. Mean of 18 trials and 2 seeds.

| | | online trials | | |
|---|---|---|---|---|
| | Variation | 0 | 10 | 20 |
| Reach | distractor | $22_{\pm 0}$ | $22_{\pm 11}$ | $\mathbf{62_{\pm 6}}$ |
| | object shape | $44_{\pm 11}$ | $44_{\pm 0}$ | $\mathbf{78_{\pm 11}}$ |
| Pick | distractors | $22_{\pm 11}$ | $56_{\pm 0}$ | $\mathbf{67_{\pm 11}}$ |
| | object color | $0_{\pm 0}$ | $0_{\pm 0}$ | $\mathbf{0_{\pm 0}}$ |
| Kitchen | distractor | $0_{\pm 0}$ | $50_{\pm 28}$ | $\mathbf{67_{\pm 11}}$ |

not only benefit from better $Q$-value estimation, but also leverage the standard deviation of the $Q$-values to measure the uncertainty of a state-action pair, i.e., whether it is out-of-distribution or not. This serves as a test-time regularizer that equips the planner with the ability to balance exploitation and exploration without explicitly introducing conservatism in training. By penalizing the actions leading to high uncertainty, we prioritize the actions that are more likely to achieve a reliably high return. Formally, we modify the estimated return in Equation 2 of an action sequence $\mathbf{a}_{0:h}$ to be

$$\hat{\mathcal{R}} = \gamma^h \left( Q_\theta(\mathbf{z}_h, \mathbf{a}_h) - \lambda u_h \right) + \sum_{t=0}^{h-1} \gamma^t \left( R_\theta(\mathbf{z}_t, \mathbf{a}_t) - \lambda u_t \right), \quad u_t = \mathrm{std}\left( \{Q_\theta^{(i)}(\mathbf{z}_t, \mathbf{a}_t)\}_{i=1}^N \right), \quad (4)$$

where the uncertainty regularizer $u_t$ is highlighted in **red**. Here, $\mathrm{std}$ denotes the standard deviation operator, and $\lambda$ is a constant coefficient that controls the regularization strength. We use the same value of $\lambda$ for both the offline and online stages in practice, but it need not be equal.

To facilitate rapid propagation of information acquired during finetuning, we maintain *two* replay buffers, $\mathcal{B}_{\mathrm{off}}$ and $\mathcal{B}_{\mathrm{on}}$ for offline and online data, respectively, and optimize the objective in Equation 1 on mini-batches of data sampled in equal parts from $\mathcal{B}_{\mathrm{off}}$ and $\mathcal{B}_{\mathrm{on}}$, *i.e.*, online interaction data is heavily oversampled early in finetuning. Balanced sampling has been explored in various settings [33, 34, 35, 22, 36, 37], and we find that it consistently improves finetuning of world models as well.

## 4 Experiments & Discussion

We evaluate our method on diverse continuous control tasks from the D4RL [38] and xArm [39] task suites and quadruped locomotion in simulation, as well as three visuo-motor control tasks on a real xArm robot, as visualized in Figure 3. Our experiments aim to answer the following questions:

— **Q1:** *How does our approach compare to state-of-the-art methods for offline RL and online RL?*
— **Q2:** *Can our approach be used to finetune world models on unseen tasks and task variations?*
— **Q3:** *How do individual components of our approach contribute to its success?*

*Table 3*. **Offline-to-online results in simulation.** Success rate (xArm) and normalized return (D4RL and quadruped) of methods **before** and **after** online finetuning. See Appendix B for task explanations. Mean of 5 seeds.

| Task | online | | offline-to-online | | |
|---|---|---|---|---|---|
| | TD-MPC | TD-MPC (+d) | TD-MPC (+o) | IQL | **Ours** |
| Push (m) | 69.0 | 76.0 | 14.0 → 77.0 | **39.0** → 51.0 | 35.0 → **79.0** |
| Push (mr) | 69.0 | **81.0** | 59.0 → 80.0 | 22.0 → 22.0 | 45.0 → 64.0 |
| Pick (mr) | 0.0 | 84.0 | 0.0 → **96.0** | 0.0 → 0.0 | 0.0 → 88.0 |
| Pick (m) | 0.0 | 52.0 | 0.0 → 58.0 | 0.0 → 1.0 | 0.0 → **66.0** |
| Hopper (m) | 4.8 | 2.8 | 0.8 → 11.0 | **66.3** → 76.1 | 49.6 → **100.7** |
| Hopper (mr) | 4.8 | 6.1 | 13.4 → 13.7 | 76.5 → **101.4** | 84.4 → 93.5 |
| AntMaze (mp) | 0.0 | 52.0 | 0.0 → 68.0 | 54.0 → 80.0 | 58.0 → **96.0** |
| AntMaze (md) | 0.0 | 72.0 | 0.0 → **91.0** | 62.0 → 88.0 | 75.0 → 89.0 |
| Walk | 8.8 | 9.2 | 18.5 → 1.2 | 19.1 → 19.2 | 67.2 → **85.8** |
| Average | 17.4 | 48.3 | 11.8 → 55.1 | 37.7 → 48.7 | **46.0 → 84.7** |

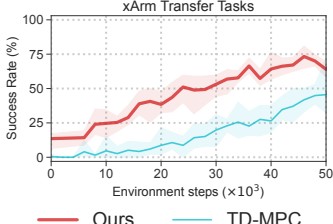

*Figure 4*. **Finetuning to unseen tasks.** Success rate (%) aggregated across 9 transfer tasks in simulated xArm environments. Mean of 5 seeds.

In the following, we first detail our experimental setup, and then proceed to address each of the above questions based on our experimental results.

**Real robot setup** Our setup is shown in Figure 3 *(right)*. The agent controls an xArm 7 robot with a jaw gripper using positional control, and $224 \times 224$ RGB image observations are captured by a static third-person Intel RealSense camera (an additional top-view camera is used for *kitchen*; see Appendix B.1 for details); the agent also has access to robot proprioceptive information. Our setup requires no further instrumentation. We consider three tasks: *reach*, *pick*, and *kitchen*, and several task variations derived from them. Our tasks are visualized in Figure 3. The goal in *reach* is to reach a target with the end-effector, the goal in *pick* is to pick up and lift a target object above a height threshold, and the goal in *kitchen* is to grasp and put a pot in a sink. We use manually designed detectors to determine task success and automatically provide sparse rewards (albeit noisy) for both offline and online RL. We use 120 offline trajectories for *reach*, 200 for *pick*, and 216 for *kitchen*; see Section 4.1 *(Q3.2)* for details and ablations.

**Simulation tasks and datasets** We consider a diverse set of tasks and datasets, including four tasks from the D4RL [38] benchmark (*Hopper (medium)*, *Hopper (medium-replay)*, *AntMaze (medium-play)* and *AntMaze (medium-diverse)*), two visuo-motor control tasks from the xArm [40] benchmark (*push and pick*) and a quadruped locomotion task (*Walk*); the two xArm tasks are similar to our real-world tasks except using lower image resolution ($84 \times 84$) and dense rewards. See Figure 3 *(left)* for task visualizations. We also consider two dataset variations for each xArm task: *medium*, which contains 40k transitions (800 trajectories) sampled from a suboptimal agent, and *medium-replay*, which contains the first 40k transitions (800 trajectories) from the replay buffer of training a TD-MPC agent from scratch. See Appendix B for more details.

**Baselines** We compare our approach against strong online RL and offline RL methods: *(i)* **TD-MPC** [26] trained from scratch with online interaction only, *(ii)* **TD-MPC (+data)** which utilizes the offline data by appending them to the replay buffer but is still trained online only, *(iii)* **TD-MPC (+offline)** which naïvely pretrains on offline data and is then finetuned online, but without any of our additional contributions, and *(iv)* **IQL** [19], a *state-of-the-art* offline RL algorithm which has strong offline performance and also allows for policy improvement with online finetuning. See Appendix C and D for extensive implementation details on our method and baselines, respectively.

## 4.1 Results

**Q1:** **Offline-to-online RL** We benchmark methods across all tasks considered; real robot results are shown in Table 1, and simulation results are shown in Table 3. We also provide aggregate curves in Figure 5 *(top)* and per-task curves in Appendix A. Our approach consistently achieves strong zero-shot and online finetuning performance across tasks, outperforming offline-to-online TD-MPC and IQL **by a large margin in both simulation and real** in terms of asymptotic performance. Notably, the performance of our method is more robust to variations in dataset and task than baselines, as evidenced by the aggregate results.

**Q2: Finetuning to unseen tasks** A key motivation for developing learning-based (and model-based in particular) methods for robotics is the potential for generalization and fast adaptation to unseen scenarios. To assess the versatility of our approach, we conduct additional offline-to-online finetuning experiments where the offline and online tasks are distinct, *e.g.*, transferring a *reach* policy to a *push* task, introducing distractors, or changing the target object. We design 5 real-world transfer tasks, and 11 in simulation (9 for xArm and 2 for locomotion). See Figure 3 *(center)* and Appendix B for task visualizations and descriptions. Our real robot results are shown in Table 2, and aggregated simulation results are shown in Figure 4. We also provide per-task results in Appendix A. Our method successfully adapts to unseen tasks and task variations – both in simulation and in real – and significantly outperforms TD-MPC trained from scratch on the target task. However, as evidenced in the *object color* experiment of Table 2, transfer might not succeed if the initial model does not achieve *any* reward.

**Q3: Ablations** To understand how individual components contribute to the success of our method, we conduct a series of ablation studies that exhaustively ablate each design choice. We highlight our key findings based on aggregate task results but note that per-task curves are available in Appendix A.

**Q3.1: Algorithmic components** We ablate each component of our proposed method in Figure 5 *(bottom)*, averaged across all D4RL tasks. Specifically, we ablate *(1)* learning $Q_\theta$ with in-sample actions and expectile regression as described in Section 3.1, *(2)* using an ensemble of 5 value functions instead of 2 as in the original TD-MPC, *(3)* regularizing planning with our uncertainty penalty described in Section 3.2, and *(4)* using balanced sampling, *i.e.*, sampling offline and online data in equal parts within each mini-batch. The results highlight the effectiveness of our key contributions. Balanced sampling improves the data-efficiency of online finetuning, and all other components contribute to both offline and online performance.

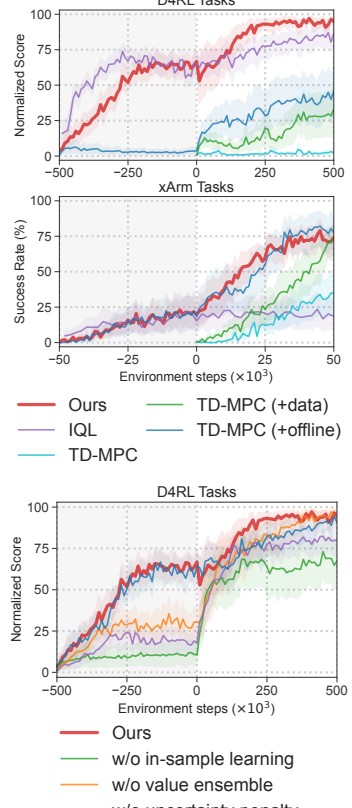

*Figure 5.* **Aggregate results.** *Top:* comparison to baselines. *Bottom:* ablations. Offline pretraining is shaded gray. 5 seeds.

**Q3.2: Offline dataset** Next, we investigate how the quantity and source of the offline dataset affect the success of online finetuning. We choose to conduct this ablation with real-world data to ensure that our conclusions generalize to realistic robot learning scenarios. We experiment with two data sources and two dataset sizes: *Base* that consists of 50/100 trajectories (depending on the dataset size) generated by a BC policy with added noise; and *Diverse* that consists of the same 50/100 tra-

*Table 4.* **Real-world ablation on offline data.** Success rate (%) as a function of online finetuning trials for two data sources and sizes from our real-world reach task. Mean of 18 trials and 2 seeds.

| | TD-MPC | | Ours | | | |
|---|---|---|---|---|---|---|
| | Base | Diverse | Base | | Diverse | |
| Trials | 100 | 120 | 50 | 100 | 70 | 120 |
| 0 | $0\pm0$ | $50\pm18$ | $0\pm0$ | $0\pm0$ | $33\pm0$ | $\mathbf{72\pm6}$ |
| 10 | $44\pm12$ | $67\pm12$ | $50\pm0$ | $61\pm6$ | $61\pm6$ | $\mathbf{94\pm6}$ |
| 20 | $\mathbf{83\pm6}$ | $78\pm12$ | $61\pm6$ | $\mathbf{89\pm0}$ | $\mathbf{89\pm0}$ | $\mathbf{89\pm0}$ |

jectories as Base, but with an additional 20 exploratory trajectories from a suboptimal RL agent. In fact, the additional trajectories correspond to a 20-trial replay buffer from an experiment conducted in the early stages of the research project. Results for this experiment are shown in Table 4. We find that results improve with more data regardless of source, but that exploratory data holds far greater value: neither TD-MPC nor our method succeeds zero-shot when trained on the Base dataset – regardless of data quantity – whereas our method obtains 33% and 72% success rate with 70 and 120 trajectories, respectively, from the Diverse dataset. This result demonstrates that replay buffers from previously trained agents can be valuable data sources for future experiments.

**Q3.3:** **Uncertainty regularization** We seek to understand how the uncertainty regularization strength $\lambda$ influences the test-time performance of our method. Results for two tasks are shown in Figure 6; see Appendix A for more results. While $\lambda > 0$ almost always outperforms $\lambda = 0$ in both offline and online RL, large values, *e.g.*, $\lambda = 20$, can be detrimental to online RL in some cases. We use the same $\lambda$ for offline and online stages in this work, but remark that our uncertainty regularizer is a *test-time* regularizer, which can be tuned at any time at no cost.

## 5 Related Work

**Offline RL** algorithms seek to learn RL policies solely from pre-existing datasets, which results in a state-action distribution shift between training and evaluation. This shift can be mitigated by applying explicit regularization techniques to conventional online RL algorithms, commonly Soft Actor-Critic (SAC; [41]). Most prior works constrain policy actions to be close to the data [13, 14, 15, 16], or regularize the value function to be conservative (*i.e.*, underestimating) [17, 18, 42, 43]. While these strategies can be highly effective for offline RL, they often slow down

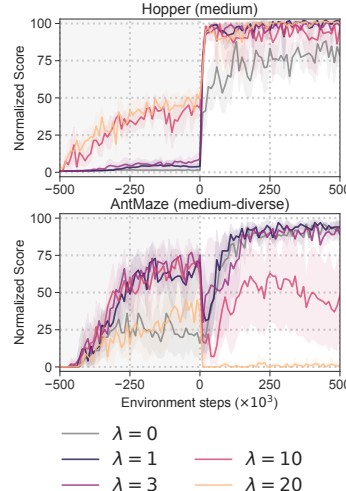

*Figure 6.* **Regularization ($\lambda$).** Score as a function of regularization strength for two tasks from D4RL. Gray shade indicates offline RL. Mean of 5 seeds.

convergence when finetuned with online interaction [21, 22, 24]. Lastly, Agarwal et al. [44] and Yarats et al. [23] show that online RL algorithms are sufficient for offline RL when data is abundant.

**Finetuning policies with RL** Multiple works have considered finetuning policies obtained by, *e.g.*, imitation learning [45, 12, 36], self-supervised learning [46], or offline RL [21, 47, 22, 48, 24]. Notably, Rajeswaran et al. [45] learns a model-free policy and constrains it to be close to a set of demonstrations, and Lee et al. [22] shows that finetuning an ensemble of model-free actor-critic agents trained with conservative $Q$-learning on an offline dataset can improve sample-efficiency in simulated control tasks. By instead learning a *world model* on offline data, our method regularizes actions at *test-time* (via planning) based on model uncertainty, *without* explicit loss terms, which is particularly beneficial for *few-shot* finetuning. While we do not compare to [22] in our experiments, we compare to IQL [19], a concurrent offline RL method that is conceptually closer to our method.

**Real-world RL** Existing work on real robot learning typically trains policies on large amounts of data in simulation, and transfers learned policies to real robots without additional training (simulation-to-real). This introduces a domain gap, for which an array of mitigation strategies have been proposed, including domain randomization [8, 4, 1], data augmentation [39], and system identification [49]. Additionally, building accurate simulation environments can be a daunting task. At present, only a limited number of studies have considered training RL policies in the real world without any reliance on simulators [50, 21, 51]. For example, researchers have proposed to accelerate online RL with human demonstrations [50] or offline datasets [21, 34] for model-free algorithms. Most recently, Wu et al. [51] demonstrates that MBRL can be data-efficient enough to learn diverse real robot tasks from scratch. Our work is conceptually similar to [21, 51] but is not directly comparable, as we consider an order of magnitude less online data than prior work.

## 6 Limitations

Several open problems remain: offline data quality and quantity heavily impact few-shot learning, and we limit ourselves to sparse rewards since dense rewards are difficult to obtain in the real world. We also find that the optimal value of $\lambda$ can differ between tasks and between offline and online RL. We leave it as future work to automate this hyperparameter search, but note that doing so is relatively cheap since it can be adjusted at test-time without any overhead. Lastly, we consider pretraining on a single task and transferring to unseen variations. Given such limited data for pretraining, some structural similarity between tasks is necessary for few-shot learning to be successful.

**Acknowledgments**

This work was supported, in part, by the Amazon Research Award, gifts from Qualcomm and the Technology Innovation Program (20018112, Development of autonomous manipulation and gripping technology using imitation learning based on visual and tactile sensing) funded by the Ministry of Trade, Industry & Energy (MOTIE), Korea.

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

# A  Additional Results

In addition to the aggregated results in the main paper, we also provide per-task results for the experiments and tasks in simulation. Our benchmark results are shown in Figure 7, and task transfer results are shown in Figure 10. Per-task results for ablations are shown in Figure 8 and Figure 9.

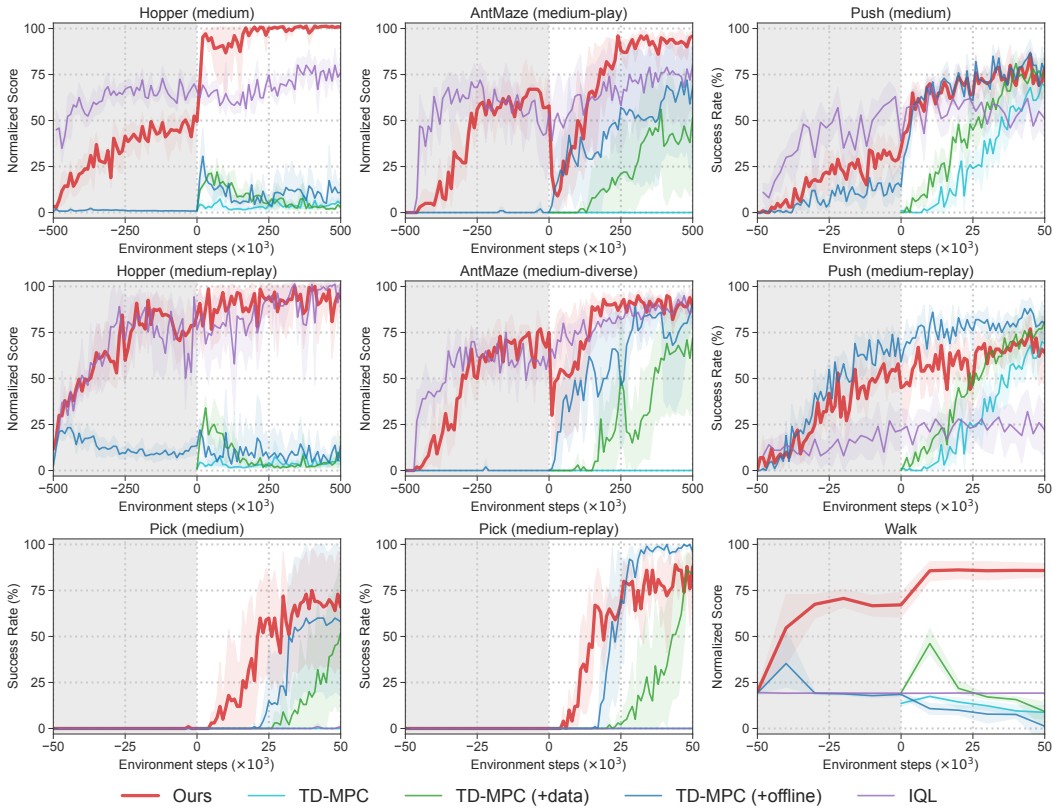

*Figure 7.* **Comparison of our method against baselines.** Offline pretraining is shaded gray. Mean of 5 seeds; shaded area indicates 95% CIs.

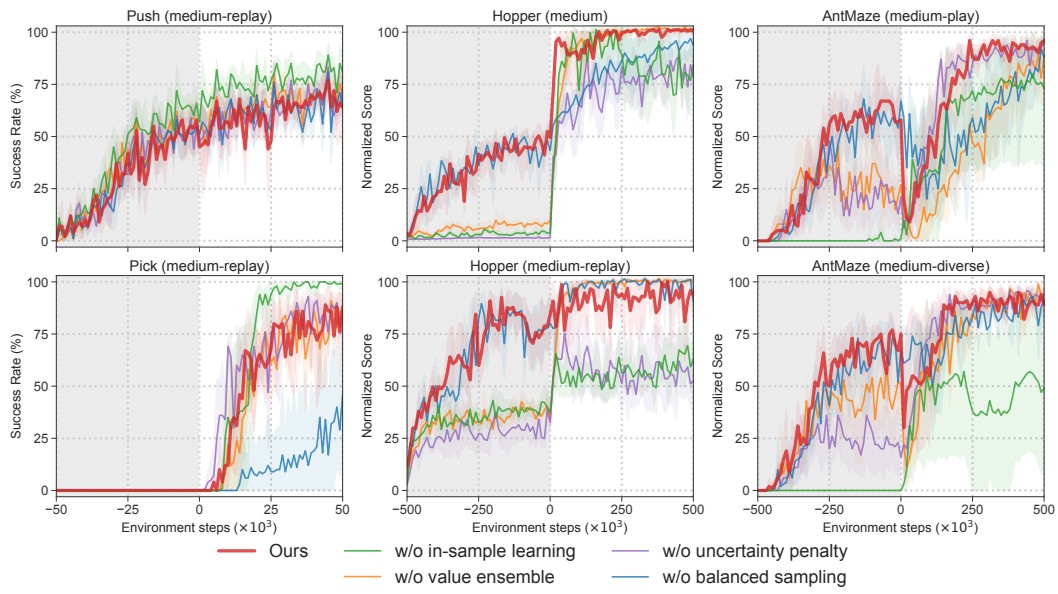

*Figure 8.* **Per-task ablation results.** Offline pretraining is shaded gray. Mean of 5 seeds; shaded area indicates 95% CIs.

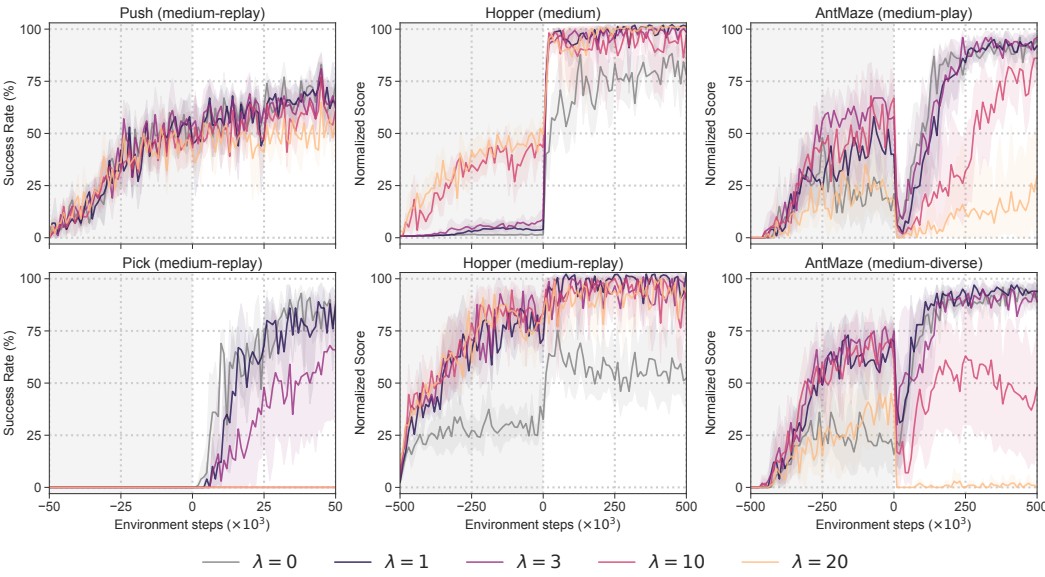

*Figure 9.* **Ablation study on uncertainty coefficient** ($\lambda$). Offline pretraining is shaded gray. Mean of 5 seeds; shaded area indicates 95% CIs.

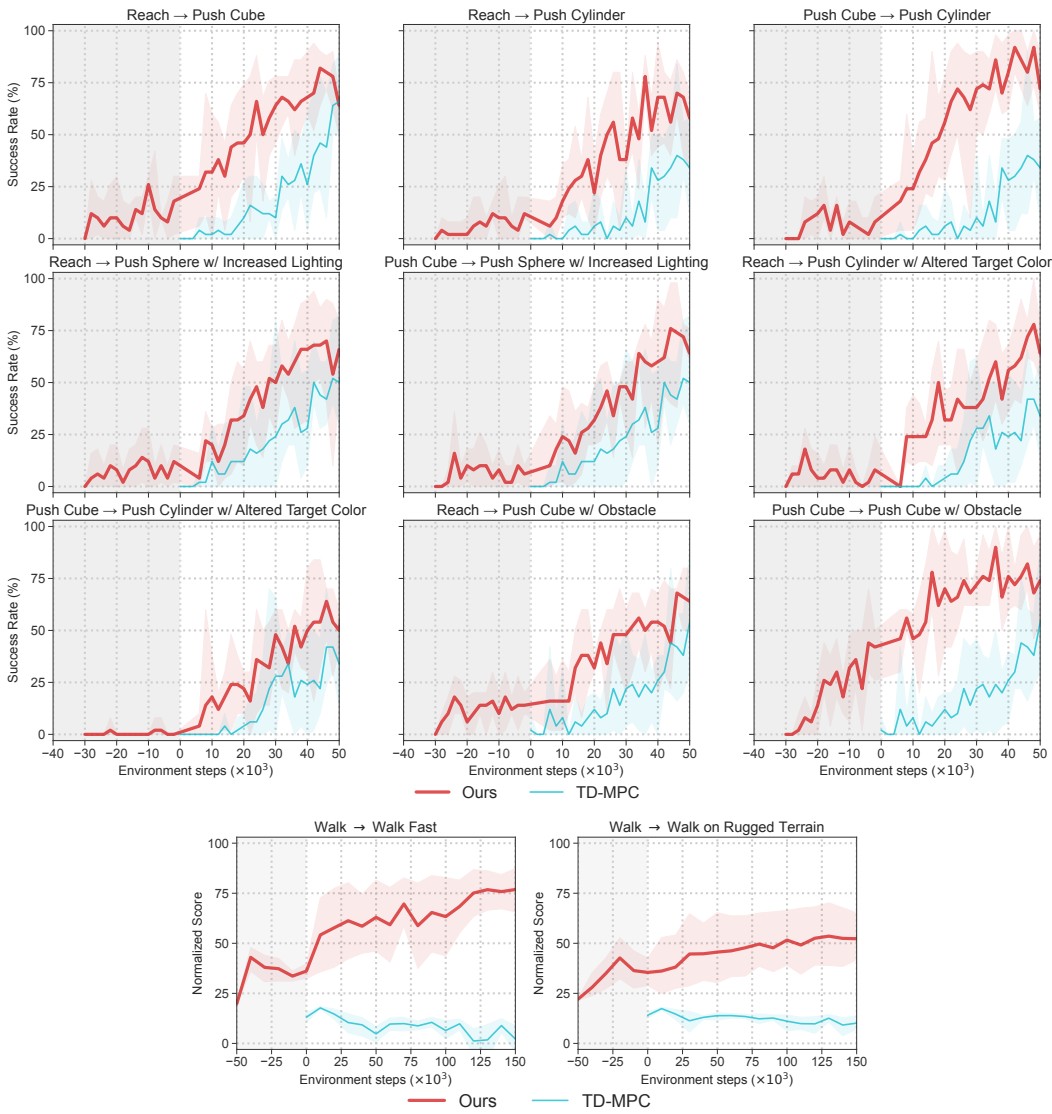

*Figure 10.* **Task transfer results.** Success rate (%) of our method and TD-MPC trained from scratch on all simulated transfer tasks. The first nine are designed based on the xArm [40] task suite, and the last two are quadruped locomotion. Offline pretraining is shaded gray. Mean of 5 seeds; shaded area indicates 95% CIs.

# B  Tasks and Datasets

## B.1  Real-World Tasks and Datasets

We implement three visuo-motor control tasks, *reach*, *pick* and *kitchen* on a UFactory xArm 7 robot arm. Here we first introduce the setup for *reach* and *pick*, which share the same workspace. We use an Intel RealSense Depth Camera D435 as the only external sensor. The observation space contains a $224 \times 224$ RGB image and an 8-dimensional robot proprioceptive state including the position, rotation, and the opening of the end-effector and a boolean value indicating whether the gripper is stuck. Both tasks are illustrated in Figure 3 *(second from the left)*. For safety reasons, we limit the moving range of the gripper in a $30cm \times 30cm \times 30cm$ cube, of which projection on the table is illustrated in Figure 11. To promote consistency between experiments, we evaluate agents on a set of fixed positions, visualized as red crosses in the aforementioned figure. The setup for *kitchen* is shown in Figure 12. We use two D435 cameras for this task, providing both a front view and a top view. The observation space thus contains two $224 \times 224$ RGB images and the 8-dimensional robot proprioceptive state.

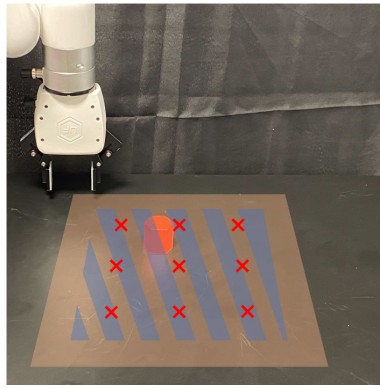

*Figure 11.* **Real-world workspace.** Moving range of the end-effector and the initialization range of target/object are shaded in the image. The positions for evaluation are labeled by crosses.

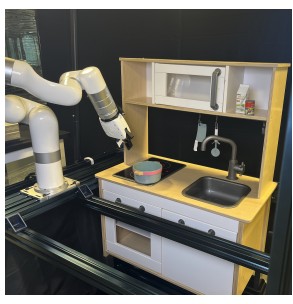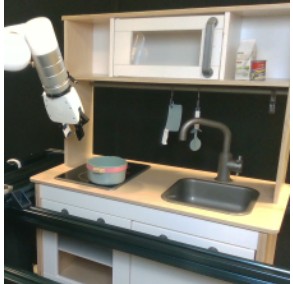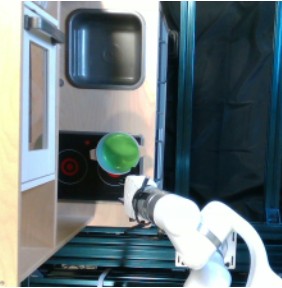

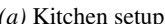

*(a)* Kitchen setup      *(b)* Front view      *(c)* Top view

*Figure 12.* **Real-world kitchen task setup.** *(a)* Setup of the kitchen workspace with the xArm robot. *(b)-(c)* Sample images from the front view and the top view, respectively.

Below we describe each task and the data used for offline pretraining in detail. Figure 13 shows sample trajectories for these tasks.

***Reach***    The objective of this task is to accurately position the red hexagonal prism, held by the gripper, above the blue square target. The action space of this task is defined by the first two dimensions, which correspond to the horizontal plane. The agent will receive a reward of 1 when the object is successfully placed above the target, and a reward of 0 otherwise. The offline dataset for *reach* comprises 100 trajectories collected using a behavior-cloning policy, which exhibits an approximate success rate of 50%. Additionally, there are 20 trajectories collected through teleoperation, where the agent moves randomly, including attempts to cross the boundaries of the allowable end-effector movement. These 20 trajectories are considered to be diverse and are utilized for conducting an ablation study around the quality of the offline dataset.

***Pick***    The objective of this task is to grasp and lift a red hexagonal prism by the gripper. The action space of this task contains the position of the end-effector and the opening of the gripper. The agent will receive a reward of 1 when the object is successfully lifted above a height threshold, 0.5 when the object is grasped but not lifted, and 0 otherwise. The offline dataset for *pick* comprises 200 trajectories collected using a BC policy that has an approximate success rate of 50%.

**Kitchen**  This task requires the xArm robot to grasp a pot and put it into a sink in a toy kitchen environment. The agent will receive a reward of 1 when the pot is successfully placed in the sink, 0.5 when the pot is grasped, and 0 otherwise. The offline dataset for *kitchen* consists of 216 trajectories, of which 100 are human teleoperation trajectories, 25 are from BC policies, and 91 are from offline RL policies.

**Real-world transfer tasks**  We designed two transfer tasks for both *reach* and *pick*, and one for *kitchen* as shown in Figure 3 *(the second from right)*. As the red hexagonal prism is an important indicator of the end-effector position in *reach*, we modify the task by (1) placing an additional red hexagonal prism on the table, alongside the existing one, and (2) replacing the object with a small red ketchup bottle, whose bottom is not aligned with the end-effector. In *pick*, the red hexagonal prism is regarded as a target object. Therefore we (1) add two distractors, each with a distinct shape and color compared to the target object, and (2) change the color and shape of the object (from a red hexagonal prism to a green octagonal prism). For *kitchen*, we also add a teapot with a similar color as the pot in the scene as a distractor. We've shown by experiments that different modifications will have different effects on subsequent performance in finetuning, which demonstrates both the effectiveness and limitation of the offline-to-online pipeline we discussed.

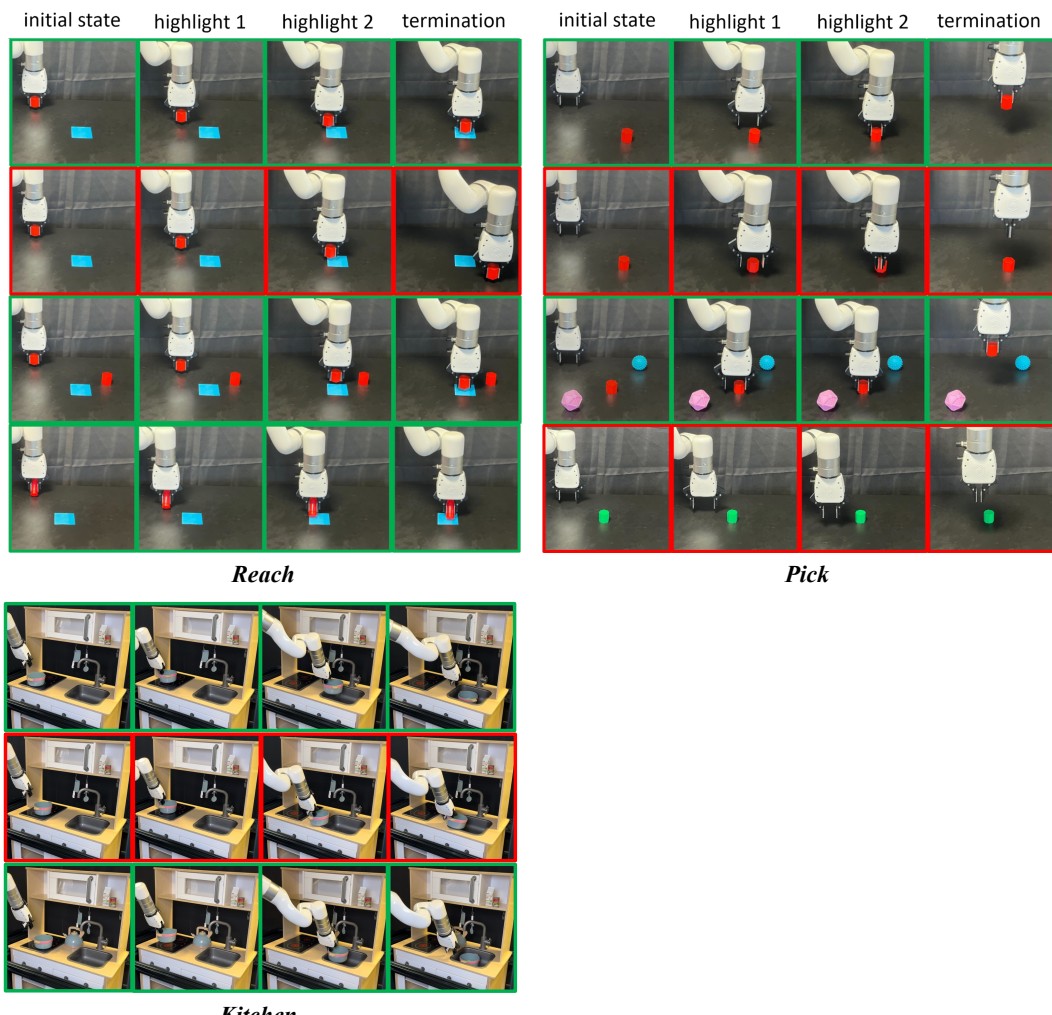

*Figure 13.* **Sample trajectories.** We include eleven trajectories from the offline dataset or evaluation results, which illustrate all real-world tasks considered in this work. Successful trajectories are marked green while failed trajectories are marked red.

## B.2 Simulation Tasks and Datasets

**xArm** *Push* and *pick* are two visuo-motor control tasks in the xArm robot simulation environment [40] implemented in MuJoCo. The observations consist of an $84 \times 84$ RGB image and a 4-dimensional robot proprioceptive state including the position of the end-effector and the opening of the gripper. The action space is the control signal for this 4-dimensional robot state. The tasks are visualized in Figure 3 *(left)*. *push* requires the robot to push a green cube to the red target. The goal in *pick* is to pick up a cube and lift it above a height threshold. Handcrafted dense rewards are used for these two tasks. We collected the offline data for offline-to-online finetuning experiments by training TD-MPC agents from scratch on these tasks. The *medium* datasets contain 40k transitions (800 trajectories) sampled from a sub-optimal agent, and the *medium-replay* datasets contain the first 40k transitions (800 trajectories) from the replay buffers. Figure 14 gives an overview of the offline data distribution for the two tasks.

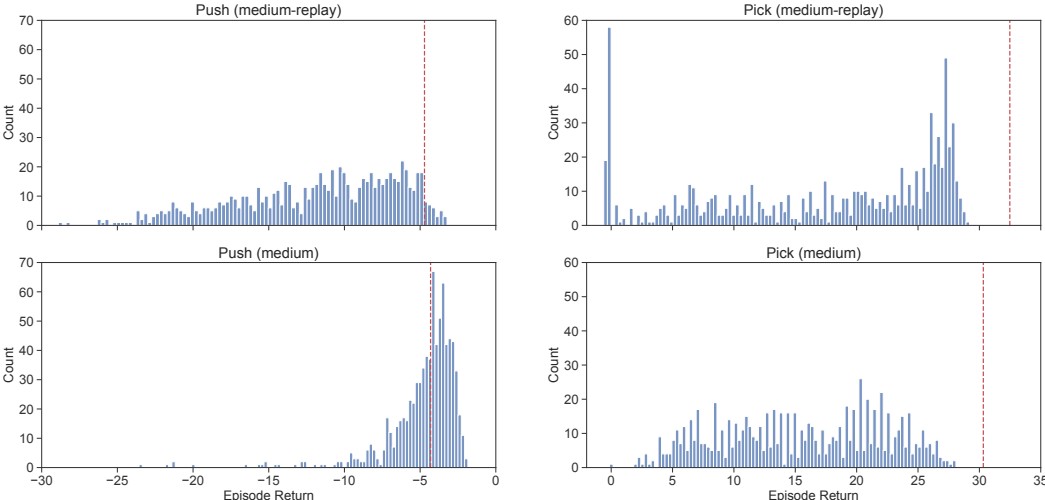

*Figure 14.* **Offline dataset statistics for xArm tasks in simulation.** We plot the distribution of episode returns for trajectories in the two offline datasets. The red line indicates the mean performance achieved by our method after online finetuning.

**Quadruped locomotion** *Walk* is a state-only continuous control task with a 12-DoF Unitree Go1 robot, as visualized in Figure 3 *(left)*. The policy takes robot states as input and output control signal for 12 joints. The goal of this task is to control the robot to walk forward at a specific velocity. Rewards consist of a major velocity reward that is maximized when the forward velocity matches the desired velocity, and a minor component that penalizes unsmooth actions.

**Transfer tasks** We designed nine transfer tasks based on *reach* (the same task as real *reach* but simplified because of the knowledge of ground-truth positions) and *push* with xArm, and two transfer tasks with the legged robot in simulation to evaluate the generalization capability of offline pretrained models. Compared to real-world tasks, the online budget is abundant in simulation, thus we increase the disparity between offline and online tasks such as finetuning on a totally different task. As the target point for both xArm tasks is a red circle, we directly use *reach* as offline pretrain task and online finetuning on different instances of *push* including push cube, push sphere, push cylinder, and push cube with an obstacle. For quadruped locomotion, we require the robot to walk at a higher target speed (twice the pretrained speed) and to walk on new rugged terrain. Tasks are illustrated in Figure 15.

**D4RL** We consider four representative tasks from two domains (Hopper and AntMaze) in the D4RL [38] benchmark. Each domain contains two data compositions. *Hopper* is a Gym locomotion domain where the goal is to make hops that move in the forward (right) direction. Observations

contain the positions and velocities of different body parts of the hopper. The action space is a 3-dimension space controlling the torques applied on the three joints of the hopper. *Hopper (medium)* uses 1M samples from a policy trained to approximately 1/3 the performance of the expert, while *Hopper (medium-replay)* uses the replay buffer of a policy trained up to the performance of the medium agent. *Antmaze* is a navigation domain with a complex 8-DoF quadruped robot. We use the *medium* maze layout, which is shown in Figure 3 *(left)*. The *play* dataset contains 1M samples generated by commanding specific hand-picked goal locations from hand-picked initial positions, and the *diverse* dataset contains 1M samples generated by commanding random goal locations in the maze and navigating the ant to them. This domain is notoriously challenging because of the need to "stitch" suboptimal trajectories. These four tasks are officially named `hopper-medium-v2`, `hopper-medium-replay-v2`, `antmaze-medium-play-v2` and `antmaze-medium-diverse-v2` in the D4RL benchmark.

## C  Implementation Details

*Q***-ensemble and uncertainty estimation**   We provide PyTorch-style pseudo-code for the implementation of the $Q$-ensemble and uncertainty estimation discussed in Section 3.2. Here `Qs` is a list of $Q$-networks. We use the minimum value of two randomly selected $Q$-networks for $Q$-value estimation, and the uncertainty is estimated by the standard deviation of all $Q$-values. We use five $Q$-networks in our implementation.

```python
def Q_estimate(Qs, z, a):
    x = torch.cat([z, a], dim=-1) # concatenate (latent) state and action
    idxs = random_choice(len(Qs), 2, replace=False) # randomly select two distinct Qs
    q1, q2 = Qs[idxs[0]](x), Qs[idxs[1]](x)
    return torch.min(q1, q2)  # return the minimum of the two as Q value estimation

def Q_uncertainty(Qs, z, a):
    x = torch.cat([z, a], dim=-1) # concatenate (latent) state and action
    qs = torch.stack(list(q(x) for q in Qs), dim=0)
    uncertainty = qs.std(dim=0)   # compute the standard deviation as uncertainty
    return uncertainty
```

**Network architecture**   For the real robot tasks and simulated xArm tasks where observations contain both an RGB image and a robot proprioceptive state, we separately embed them into feature vectors of the same dimensions with a convolutional neural network and a 2-layer MLP respectively, and do element-wise addition to get a fused feature vector. For real-world kitchen tasks, where observations include two RGB images and a proprioceptive state, we use separate encoders to embed them into three feature vectors and do the element-wise addition. For D4RL and quadruped locomotion tasks where observations are state features, only the state encoder is used. We use five $Q$-networks to implement the $Q$-ensemble for uncertainty estimation. All $Q$-networks have the same architecture. An additional $V$ network is used for state value estimation as discussed in Section 3.1.

**Hyperparameters**   We list the hyperparameters of our algorithm in Table 5. The hyperparameters related to our key contributions are highlighted .

**Other details**   We apply image shift augmentation [52] to image observations, and use Prioritized Experience Replay (PER; [53]) when sampling from replay buffers.

## D  Baselines

**TD-MPC**   We use the same architecture and hyperparameters for our method and our three TD-MPC baselines as in the public TD-MPC implementation from https://github.com/nicklashansen/tdmpc, except that multiple encoders are used to accommodate both visual inputs and robot proprioceptive information in the real robot and xArm tasks, as described in Apppendix C.

For the **TD-MPC (+data)** baseline, we append the offline data to the replay buffer at the beginning of online training so that they can be sampled together with the newly-collected data for model update. For the **TD-MPC (+offline)** baseline, we naïvely pretrain the model on offline data and then finetune it with online RL without any changes to hyperparameters.

**IQL**  We use the official implementation from `https://github.com/ikostrikov/implicit_q_learning` for the IQL baseline. We use the same hyperparameters that the authors used for D4RL tasks. For xArm tasks, we perform a grid search over the hyperparameters $\tau \in \{0.5, 0.6, 0.7, 0.8, 0.9, 0.95\}$ and $\beta \in \{0.5, 1.0, 3.0, 10.0\}$, and we find that expectile $\tau = 0.95$ and temperature $\beta = 10.0$ achieves the best results. We add the same image encoder as ours to the IQL implementation in visuo-motor control tasks.

*Table 5.* **Hyperparameters.**

| Hyperparameter | Value |
|---|---|
| Expectile ($\tau$) | 0.9 (AntMaze, xArm, Walk) |
| | 0.7 (Hopper) |
| AWR temperature ($\beta$) | 10.0 (AntMaze) |
| | 3.0 (Hopper, xArm) |
| | 1.0 (Walk) |
| Uncertainty coefficient ($\lambda$) | 1 (xArm, Walk) |
| | 3 (AntMaze) |
| | 20 (Hopper) |
| $Q$ ensemble size | 5 |
| Batch size | 256 |
| Learning rate | 3e-4 |
| Optimizer | Adam($\beta_1 = 0.9, \beta_2 = 0.999$) |
| Discount | 0.99 (D4RL, Walk) |
| | 0.9 (xArm) |
| Action repeat | 1 (D4RL, Walk) |
| | 2 (xArm) |
| Value loss coefficient | 0.1 |
| Reward loss coefficient | 0.5 |
| Latent dynamics loss coefficient | 20 |
| Temporal coefficient | 0.5 |
| Target network update frequency | 2 |
| Polyak | 0.99 |
| MLP hidden size | 512 |
| Latent state dimension | 50 |
| Population size | 512 |
| Elite fraction | 50 |
| Policy fraction | 0.1 |
| Planning iterations | 6 (xArm, Walk) |
| | 1 (D4RL) |
| Planning horizon | 5 |
| Planning temperature | 0.5 |
| Planning momentum coefficient | 0.1 |

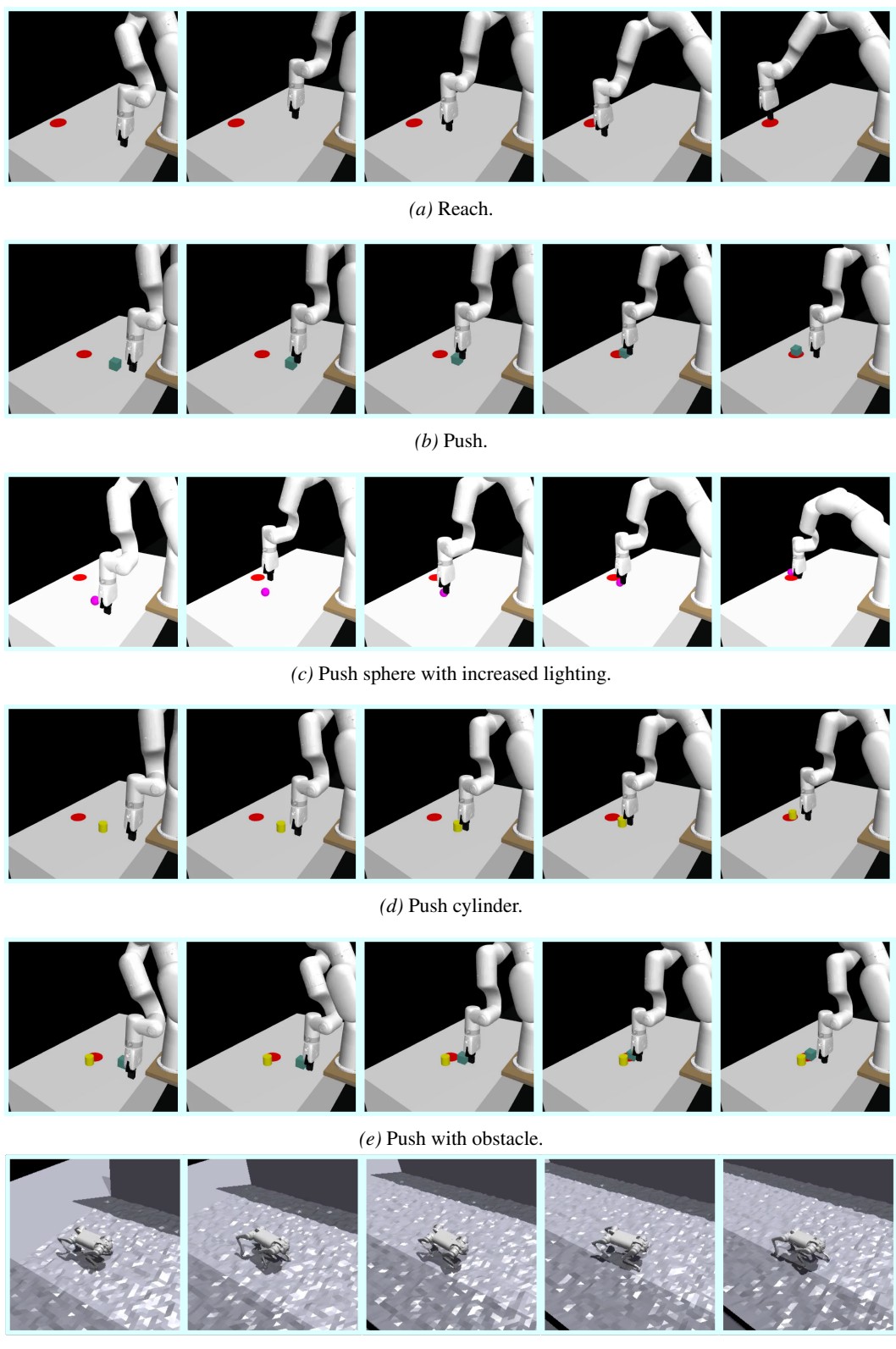

*(a)* Reach.

*(b)* Push.

*(c)* Push sphere with increased lighting.

*(d)* Push cylinder.

*(e)* Push with obstacle.

*(f)* Walk on rugged terrain.

*Figure 15.* **Transfer tasks in simulation.** We consider a total of eleven transfer settings in simulation. We here visualize a trajectory for each of the tasks used in our xArm experiments, and a trajectory for walking on rugged terrain with the legged robot. Task labels correspond to those shown in Figure 10.

