# OpenReview forum: "Finetuning Offline World Models in the Real World"
_robot-learning.org/CoRL/2023/Conference — CoRL 2023 Oral_

### Official Review · Reviewer_UN8h · 2023-07-18

**Confidence:** 4
**Originality:** Good
**Technical Quality:** Very Good
**Clarity Of Presentation:** Very Good
**Impact:** 3

**Recommendation:**

Weak Accept: I recommend accepting the paper, but will not argue for my recommendation if the majority of other reviewers have a different opinion.

**Review:**

This paper was very well presented and well written. In general the idea of training on a dataset of previous experience then fine tuning on the robot is very appealing, especially for a small amount of new data and considering model uncertainty at test time. The results show improvements over existing methods on the real robot. The ablations effectively highlight the importance of each improvement. It was good to see that previously collected real-world data improved zero shot transfer (Q3.2), but also good to see similar performance across datasets with more interaction.

“Our approach consistently achieves strong zero-shot and online finetuning performance across all tasks, outperforming offline-to-online TD-MPC and IQL by a large margin in both simulation and real in terms of asymptotic performance.” This just doesn’t seem true looking at the aggregate results. For the zero-shot (environment steps = 0?) it doesn’t seem to outperform any method. For the xArm in simulation it does worse than TD-MPC and TD-MPC (+offline). It seems to do somewhat better on the D4RL tasks, but from appendix A, it looks like the Hopper is doing most of the heavy lifting. The real world results are good to see.

A significant claim was the uncertainty penalty could be used as a regularizer to improve actions at test-time. I feel like I’m missing something, the results only seem to report what is happening during training (offline then online). Where are the test-time results showing this improvement? Q3.3 shows the effect of parameter λ, but this doesn’t seem to be at test time.

**Quality Of The Limitations Section:**

Limitations are addressed clearly

**Questions For Rebuttal:**

In the transfer tasks, Figure 4 and Figure 10, TD-MPC is trained from scratch whereas the method from this paper is trained on offline data first (from a different task?). These don’t seem like a fair comparison. Is there a reason we don’t see TD-MPC (+offline) for the same offline datasets?

What is a trial? Is this a fixed number of environment steps or termination?

How is the perception encoder trained (h ϕ)?

Double check references include the conference citation if accepted, not just the preprint:
E.g:
Video PreTraining (VPT): Learning to Act by Watching Unlabeled Online Videos -> NeurIPS 22
MOReL: Model-Based Offline Reinforcement Learning -> NeurIPS 2020



**Robotics Focus:**

Sufficient demonstration on hardware

**Summary Of Paper:**

This work presents offline training (fixed data) then online fine-tuning in several simulation tasks (D4RL, xArm), as well as real world experiments with an xArm performing pick and place. This paper introduces several improvements to TD-MPC [1]. The first is to include in-sampling of actions (from IQL [2]) to reduce overestimation errors in value function prediction. Secondly, an ensemble of Q functions are employed, both to improve Q function predictions, and to utilise the standard deviation of the ensembles as an uncertainty penalty added to the reward and the terminal Q value. One claim is that the uncertainty penalty can be used as a regularizer at test-time, retaining model expressiveness through training.

[1] N. Hansen, X. Wang, and H. Su. Temporal difference learning for model predictive control. In
ICML, 2022.

[2] Kostrikov, A. Nair, and S. Levine. Offline reinforcement learning with implicit q-learning.
ICLR 2022


**Summary Of Recommendation:**

While in general I like the ideas introduced (bringing together of several previous works), the results don't seem overly significant. The real world experiments are promising.

After considering the additional information provided in the rebuttal I have increased my recommendation to a weak accept. Thank you to the authors for their efforts.

---

### Official Review · Reviewer_LZqg · 2023-07-18

**Confidence:** 3
**Originality:** Good
**Technical Quality:** Very Good
**Clarity Of Presentation:** Very Good
**Impact:** 3

**Recommendation:**

Weak Accept: I recommend accepting the paper, but will not argue for my recommendation if the majority of other reviewers have a different opinion.

**Review:**

In general, I like the stated goals and aim of this paper - it makes sense to use offline datasets and model-based planning to solve robotic tasks in a more data-efficient manner (especially compared to model-free RL). Additionally, while not in themselves novel, the adjustments the authors make to the TD-MPC approach seem sound, and make sense in the offline-to-online fine-tuning setting the paper focuses on. The authors provide a set of experiments which covers multiple domains with a variety of dynamics and reward structures, and show that their method is at least competitive with existing approaches.

The paper is nicely organized and well written, and presents its claims clearly. Figures and tables in the paper are also clears and easy to understand.


While the results presented in the paper seem promising, some concerns arise regarding the claims of “outperforming SoTA methods”.
One such concern is that the aggregated result plots provided in the main text may be somewhat misleading. As evident in the appendix (Figure 7), In both xArm tasks, it seems like the proposed method does not outperform the TD-MPC baseline; in most D4RL tasks, it seems like the proposed method is comparable to IQL, but does not significantly outperform it. As IQL was tuned specifically for the D4RL tasks, this raises concern that with proper hyperparameter tuning, IQL may outperform the method proposed in this paper on the xArm tasks as well.
This issue is also apparent in Table 3: the proposed method only outperforms both baselines on two specific simulated tasks in the offline-to-online setting. One wonders whether the baselines would perform better if their hyperparameters were tuned to the task they were not originally trained on (i.e. TD-MPC to the D4RL tasks, and IQL to the xArm tasks).

Regarding the claims of adjustment to “unseen tasks”:
If a Q-value estimate is learned from the offline data, is this method only relevant when applied to tasks with similar reward structures? The method claims fine tuning to unseen tasks - however, this may be misleading if it only refers to unseen distractors in similar tasks. One of the main advantages of planning using world models is that they can be used for multiple tasks, regardless of the task objective (given by the reward); in this work, with the reward model learned along with the dynamics, this advantage is forfeit.
This design of the world model to include the value function may explain its success on the “unseen” tasks in the experiments - these are all pretty close to the training tasks, with rewards similar and only visual appearances of objects or background changing. While robustness to distractors is an important trait in world models for robots, it seems like this property is also only available when visual features do not change too much - the failure case of picking up the green object instead of the red one seems like one that a model should be able to solve with appropriate fine-tuning.

In order to cement a more robust demonstration of the claims made in the introduction, the experiments should include some scenarios where the reward changes significantly between the "seen" and "unseen" tasks. At the very least, this should be addressed in the limitations section.

**Quality Of The Limitations Section:**

Limitations are addressed clearly

**Questions For Rebuttal:**

Looking at the per-task ablation study, seems like removing in-sample learning improves performance on the xArm tasks. Any intuition on why this happens?

In general, the ‘pick’ task seems to stand out, with the offline training not contributing to the fine-tuning process, and the uncertainty regularization hurting, rather than improving, performance. Is the offline data distribution here extremely different from the online distribution?

Were ablations performed on the real-world tasks and experiments? Would be interesting to see how TD-MPC performs on the offline to online real world tasks (Table 1) with only some of the improvements proposed in this paper.

Line 86: Transition function T is for full (unknown) states s, as mentioned on line 71. How come the hidden states z also obey it?

**Robotics Focus:**

Sufficient demonstration on hardware

**Summary Of Paper:**

This paper suggests a set of adjustments to the TD-MPC model-based RL algorithm, and applies them to an offline-to-online setting applicable to robotic tasks. In particular, the paper suggests a regularization technique based on an estimate of the epistemic uncertainty of the dynamics model, obtained by training an ensemble of Q-value networks. This uncertainty weighting helps limit the effect of out-of-distribution transitions when fine-tuning a model trained on an offline dataset.

**Summary Of Recommendation:**

While the additions made to the TD-MPC method seem sound and adequate for the scenarios discussed in the paper, they are not in themselves novel our groundbreaking.The results demonstrated on the various experiments are not as significantly dominant over baselines as claimed by the authors. While the paper describes an interesting method and the initial results do seem promising for fine-tuning models trained on offline data, further experimentation may be required in order for the results to be convincing.

EDIT: after reading through the other reviews and considering the information and experiments added in the rebuttal, including the updated limitations section, I am inclined towards accepting this paper and have updated my score to "weak accept" accordingly. While I am still not fully convinced of the SoTA claims for the results, they do back the authors claims on the usability of their method; I believe this paper is a valuable addition to the robot learning community.

---

### Official Review · Reviewer_mpkG · 2023-07-20

**Confidence:** 3
**Originality:** Good
**Technical Quality:** Good
**Clarity Of Presentation:** Very Good
**Impact:** 4

**Recommendation:**

Strong Accept: I recommend accepting the paper and will argue for my recommendation even if other reviewers hold a different opinion.

**Review:**

Strengths:
* The paper is very well written and structured. It was an enjoyable read!
* The topic of using epistemic uncertainty to modulate online RL is timely and interesting. The problem is well-motivated and the literature review does a good job of contextualizing the paper in prior work.
* The authors do their due dilligence in ablating the approach and conduct many real-world robot experiments.
* I appreciate the candor and honesty with which the results are presented (both good and bad).
* Further details and experiments in the appendix provide additional context for the method and the claims made in the main paper.

Weaknesses:

* The statement "We are, to the best of our knowledge, the first work to investigate offline-to-online finetuning with MBRL on real robots" is bold. From a quick search, I could not find a paper that does this exactly, but I would make sure to do a thorough literature review to ensure the accuracy of the statement. The literature review is also missing, for example, the following works:

[A] A. Kumar et al. "Pre-training for robots: Offline RL enables learning new tasks from a handful of trials." arXiv, 2022.
[B] S. M. Richards et al. "Adaptive-Control-Oriented Meta-Learning for Nonlinear Systems." RSS, 2021.

Separately, I would also recommend including in the literature review the use of epistemic uncertainty estimation for RL methods. For example, the following paper:

[C] B. Charpentier et al. "Disentangling epistemic and aleatoric uncertainty in reinforcement learning." arXiv, 2022.

* My biggest concern about the paper is the presented results. The proposed method does not always outperform the baselines (e.g., in Table 3 - simulation, the proposed method performs the best in 2 of 6 tasks, but outperforms overall across tasks, in Fig. 5, from the first two plots, the method is not clearly better). The robot tasks are also pretty simple (e.g., reach and pick). That being said, I do agree that the method appears to be more robust than the baselines to task variations and new settings.

* The references have some consistency issues. For example, conference acronyms are sometimes included (e.g., ICLR in [10]) and sometimes not (e.g., [11]). When possible, the conference venue should be cited instead of ArXiv. The references should in general be proofread (e.g., should capitalize "RT-1" in [7], "I" in [11], "Q" in [18], "RL" in [23], etc.).

**Quality Of The Limitations Section:**

Limitations are addressed clearly

**Questions For Rebuttal:**

In addition to the weaknesses listed above, I am hoping the authors can address the following clarifications during the rebuttal period.
1. The authors state in the second paragraph of the paper that extrapolation errors are less of an issue for online RL because they can learn from negative rewards. However, does this not come with safety concerns?
2. Does the regularizer get used online and offline? I did not understand this point.
3. Could you clarify why TD-MPC underperforms the proposed method in Q2? If I am understanding correctly, the proposed method is offline trained on task A and online finetuned on task B, while TD-MPC is just trained on task B. I am surprised that TD-MPC does worse on task B.
4. Does the original TD-MPC paper also have an ensemble, albeit smaller than the one considered here? What are the key differences in how the ensembles are used?

**Robotics Focus:**

Sufficient demonstration on hardware

**Summary Of Paper:**

The paper presents an approach to offline-online approach for MBRL. The authors enable generalization in the presence of distribution shift from offline to online data by regularizing the policy using uncertainty on the value function as obtained from an ensemble. The proposed method is evaluated against IQL and TD-MPC baselines on both simulated and real-world robot data.

**Summary Of Recommendation:**

Overall, the paper is well written and an enjoyable, educational read. The use of epistemic uncertainty to regularize online RL is a good idea. The real-world experiments that show robust generalization to new settings are compelling. My biggest concerns with the paper center around the per-task performance in some of the experiments - where the baselines sometimes appear to do better. Also, the literature review should really cover epistemic uncertainty in RL. As it stands, I am still leaning towards accepting the paper assuming the literature review is made more complete.

---

### Official Review · Reviewer_S26C · 2023-07-23

**Confidence:** 3
**Originality:** Very Good
**Technical Quality:** Very Good
**Clarity Of Presentation:** Very Good
**Impact:** 3

**Recommendation:**

Strong Accept: I recommend accepting the paper and will argue for my recommendation even if other reviewers hold a different opinion.

**Review:**

- The paper is well-structured and easy to read.
- The proposed test-time behavior regularization in this study is simple yet experimentally shown to be highly effective. Its applicability to unknown tasks with only a few interactions in the real world, without relying on simulators, makes it particularly valuable.
- The study demonstrates high credibility by conducting experiments not only with simulations but also with real robots, comparing state-of-the-art methods for offline RL and online RL. Additionally, the credibility is further reinforced by testing the proposed approach with variations in unseen tasks and changes in hyperparameters.

**Quality Of The Limitations Section:**

Limitations are addressed clearly

**Questions For Rebuttal:**

- In the paper, the authors state that "However, as evidenced in the object color experiment of Table 2, transfer might not succeed if the initial model does not achieve any reward." Could the authors try increasing the number of trials to see if transfer becomes possible? If it becomes possible, how many trials would be needed to achieve it? I am concerned whether this issue represents an inherent limitation of the proposed method, and if it is a limitation, I would like to know why.

**Robotics Focus:**

Sufficient demonstration on hardware

**Summary Of Paper:**

In this research, the authors propose a method to address the problem of inefficient data efficiency and challenging training of conventional reinforcement learning in real robots. They achieve this by pre-training a world model using offline data and then fine-tuning the model with online data, enabling effective policy learning for new tasks. Experiments on simulators and real robots demonstrate that the approach is capable of effective learning for both seen and unseen tasks even with limited offline data.

**Summary Of Recommendation:**

This paper demonstrates high performance with a small number of trials on a real robot, and I believe it is highly effective and important for future world model research. Therefore, I recommend accepting this paper for publication.

---

### Author Response · Authors · 2023-08-13
**General Response to Reviewers and AC**

We thank all reviewers for their thoughtful comments and feedback. A revised version of our paper is attached to the response to each reviewer, where various comments have been addressed; changes are highlighted in blue. We seek to address two common concerns here.

1. **Performance compared to baselines**. Firstly we want to point out that the distribution/quality of offline data plays an important role in algorithm performance, which has been discussed in Q3.2 as well. Recent work[1] has shown that with sufficiently diverse data, vanilla off-policy RL agents can outperform carefully designed offline RL algorithms. We want to highlight that our method is superior especially in scenarios where only limited and narrow data is available, which are also more common in real-world robot learning. To better showcase how our method performs on xArm tasks with narrow data, we consider another regime where the offline dataset is collected by rolling out a suboptimal (medium) policy trained by TD-MPC, instead of from a replay buffer which contains diverse exploratory trajectories. We use *medium*(m) to denote the new variation, and *medium-replay*(mr) to denote the old one where offline data is from the replay buffer. Updated results are shown in [Table 3](https://drive.google.com/file/d/10FX62vz-IrZNiaC13bijv6GpT3as_G6x/view?usp=sharing) and [Figure 7](https://drive.google.com/file/d/1Vah2Uyallc58OOGeWZGJjzCEggEd9_Xi/view?usp=sharing) (Push(medium) and Pick(medium)). We also want to highlight the difference between the two datasets for each D4RL task. IQL and our method achieve comparable performance on Hopper (medium-replay) and AntMaze (medium-diverse), where the offline data is diverse, whereas our method achieves significantly better offline-to-online performance on Hopper (medium) and AntMaze (medium-play), where the offline data has relatively narrow distribution.
To better demonstrate the capability of our method, we also include additional results on more challenging tasks:
- **Quadruped locomotion (sim)**.  This task requires a legged robot (12-DoF) to walk forward at a specific velocity. Our method is able to learn from suboptimal offline data and continue to improve during online finetuning, whereas all other baselines fail to make progress. The quantitative results are shown in [Table 3](https://drive.google.com/file/d/10FX62vz-IrZNiaC13bijv6GpT3as_G6x/view?usp=sharing) and [Figure 7](https://drive.google.com/file/d/1Vah2Uyallc58OOGeWZGJjzCEggEd9_Xi/view?usp=sharing) (LeggedWalk). A video of our method solving this task is available [here](https://drive.google.com/file/d/1ZwE9c2MzDGZFfsORQqOd1c1C4BYZ1W9f/view?usp=sharing).
- **Manipulation in kitchen (real)**. This task requires the xArm robot to put a bowl into a sink. Our method can achieve a high success rate in 20 online trials with 200 trajectories provided as an offline dataset and show the ability to quickly adapt to different initialization. The video of online finetuning for this task is attached [here](https://drive.google.com/file/d/1RD0588GSPw5OM_GX44Lm1gfSlCrvQph-/view?usp=sharing).

2. **Test-time behavior regularization**. The regularization is only applied to estimating the return of an action sequence during planning. This means it happens when (1) evaluating/testing the model to report performance and (2) interacting with the environment to collect trajectories in the online phase. However, no regularization is introduced in the objectives when training the model, which makes our method different from prior works that predominantly learn an explicitly and consistently conservative value function and/or policy, such as CQL[2] and MOPO[3]. This also means offline pretraining will end up with the same model regardless of regularization strength since it’s only involved in planning when evaluating. Besides, regularizing planning during online interaction results in cautious exploration and contributes to safety. It’s also possible to gradually decrease the regularization to achieve a better safety-exploration tradeoff.

Again, we thank the reviewers for their constructive feedback. We are happy to address any further comments from reviewers.

Best,
Authors

[1] Yarats, Denis, et al. "Don't change the algorithm, change the data: Exploratory data for offline reinforcement learning." arXiv, 2022.
[2] Kumar, Aviral, et al. "Conservative Q-learning for offline reinforcement learning." NeurIPS, 2020.
[3] Yu, Tianhe, et al. "Mopo: Model-based offline policy optimization." NeurIPS, 2020.

---

### Decision · Program_Chairs · 2023-08-30

**Decision:**

Accept (Oral)

**Comment:**

The paper studies the use of offline data to pretrain a world model that is then fine-tuned with online data, e.g. from a real robot. It uses epistemic uncertainty of the model as a regularizer for online data collection. The methods' effectiveness was demonstrated on simulated and real robots.

The reviewers assessed your submission overall positively, such that I am happy to suggest acceptance.

Please make sure that the improvement and suggestions that come up in the reviews and the rebuttal are appropriately incorporated in the final version of the paper.